# Harnessing Gasotransmitters to Combat Age-Related Oxidative Stress in Smooth Muscle and Endothelial Cells

**DOI:** 10.3390/ph18030344

**Published:** 2025-02-27

**Authors:** Constantin Munteanu, Anca Irina Galaction, Gelu Onose, Marius Turnea, Mariana Rotariu

**Affiliations:** 1Department of Biomedical Sciences, Faculty of Medical Bioengineering, University of Medicine and Pharmacy “Grigore T. Popa”, 700454 Iasi, Romania; anca.galaction@umfiasi.ro (A.I.G.); mariana.rotariu@umfiasi.ro (M.R.); 2Neuromuscular Rehabilitation Clinic Division, Clinical Emergency Hospital “Bagdasar-Arseni”, 041915 Bucharest, Romania; gelu.onose@umfcd.ro; 3Faculty of Medicine, University of Medicine and Pharmacy “Carol Davila”, 020022 Bucharest, Romania

**Keywords:** gasotransmitters, oxidative stress, vascular aging, endothelial cells, smooth muscle cells, hydrogen sulfide, nitric oxide, carbon monoxide

## Abstract

Age-related oxidative stress is a critical factor in vascular dysfunction, contributing to hypertension and atherosclerosis. Smooth muscle cells and endothelial cells are particularly susceptible to oxidative damage, which exacerbates vascular aging through cellular senescence, chronic inflammation, and arterial stiffness. Gasotransmitters—hydrogen sulfide (H_2_S), nitric oxide (NO), and carbon monoxide (CO)—are emerging as promising therapeutic agents for counteracting these processes. This review synthesizes findings from recent studies focusing on the mechanisms by which H_2_S, NO, and CO influence vascular smooth muscle and endothelial cell function. Therapeutic strategies involving exogenous gasotransmitter delivery systems and combination therapies were analyzed. H_2_S enhances mitochondrial bioenergetics, scavenges ROS, and activates antioxidant pathways. NO improves endothelial function, promotes vasodilation, and inhibits platelet aggregation. CO exhibits cytoprotective and anti-inflammatory effects by modulating heme oxygenase activity and ROS production. In preclinical studies, gasotransmitter-releasing molecules (e.g., NaHS, SNAP, CORMs) and targeted delivery systems show significant promise. Synergistic effects with lifestyle modifications and antioxidant therapies further enhance their therapeutic potential. In conclusion, gasotransmitters hold significant promise as therapeutic agents to combat age-related oxidative stress in vascular cells. Their multifaceted mechanisms and innovative delivery approaches make them potential candidates for treating vascular dysfunction and promoting healthy vascular aging. Further research is needed to translate these findings into clinical applications.

## 1. Introduction

Vascular aging is a critical factor contributing to the global burden of cardiovascular diseases (CVDs), which remain the leading cause of mortality worldwide [1,2]. Aging-induced vascular structure and function alterations, including arterial stiffening, endothelial dysfunction, and impaired vascular smooth muscle cell activity, are closely associated with oxidative stress [3]. This stress is characterized by an imbalance between reactive oxygen species (ROS) production and antioxidant defenses [4], which accelerates cellular damage, inflammation, and senescence, ultimately leading to chronic vascular dysfunction [5,6].

Oxidative stress is a primary driver of vascular aging [7]. The excessive accumulation of ROS disrupts cellular homeostasis and promotes molecular damage, inflammation, and cellular senescence in vascular tissues [8,9,10]. This redox imbalance exacerbates age-related vascular dysfunction, creating a vicious cycle that accelerates the progression of CVDs [11]. Significantly, oxidative stress not only affects vascular health but also contributes to other age-related diseases, including neurodegenerative disorders, diabetes [12], and chronic kidney disease [13], further amplifying its global impact [14].

Age-related changes significantly increase the risk of hypertension [15], atherosclerosis, and other vascular disorders [16], highlighting the urgent need for targeted therapeutic strategies to combat oxidative stress in smooth muscle and endothelial cells [17,18].

Targeting oxidative stress in smooth muscle and endothelial cells is crucial for mitigating age-related vascular diseases, as these cell types play pivotal roles in maintaining vascular homeostasis [19,20]. Endothelial cells regulate vascular tone, blood flow, and inflammatory responses, while smooth muscle cells provide structural support and contractile function essential for arterial compliance [21]. Oxidative stress disrupts these critical functions, leading to endothelial dysfunction, vascular stiffness, and chronic inflammation—hallmarks of vascular aging and precursors to diseases such as hypertension, atherosclerosis, and stroke (Figure 1) [22].

Gasotransmitters—hydrogen sulfide (H_2_S), nitric oxide (NO), and carbon monoxide (CO)—have emerged as a novel class of therapeutic agents [23] with significant potential in the treatment of age-related vascular diseases [24]. These small, endogenously produced gaseous molecules play essential roles in cellular signaling and homeostasis, influencing vascular tone, oxidative stress, inflammation, and cell survival [25]. Unlike traditional pharmacological agents, gasotransmitters act through diverse pathways and mechanisms, providing a multifaceted approach to disease management [26].

NO is a well-established vasodilator and regulator of endothelial function, critical for maintaining vascular homeostasis [27]. Its impairment, commonly seen in aging and oxidative stress, contributes to endothelial dysfunction and vascular stiffness [28]. NO-based therapies aim to restore NO bioavailability, thereby improving vasodilation and reducing the risk of hypertension and atherosclerosis [29,30].

H_2_S, initially regarded as a toxic gas, is now recognized for its potent antioxidant, anti-inflammatory, and cytoprotective properties [31]. H_2_S has been shown to scavenge ROS, enhance mitochondrial function, and reduce cellular senescence, making it a promising candidate for targeting vascular oxidative stress [32].

CO, traditionally associated with toxicity, has therapeutic potential at low concentrations due to its anti-inflammatory and anti-apoptotic effects [33]. CO protects vascular cells from oxidative damage and inflammation by modulating heme oxygenase activity and reducing ROS production [34].

Advances in gasotransmitter research have also led to the development of innovative delivery systems, such as gasotransmitter-releasing molecules [35] (e.g., Sodium Hydrogen Sulfide (NaHS) [36], S-nitroso-N-acetylpenicillamine (SNAP) [37], Carbon Monoxide-Releasing Molecules (CORMs) [38]) and nanotechnology-based approaches [39], which enable targeted and controlled administration. These therapies, combined with their pleiotropic effects, position gasotransmitters as promising tools for combating age-related oxidative stress and vascular dysfunction, heralding a new era in vascular medicine [40,41].

Given the projected increase in aging populations [42], the health and economic burden of vascular aging will likely escalate in the coming decades [43]. Addressing this challenge requires innovative therapeutic strategies that target the underlying mechanisms of oxidative stress and vascular dysfunction [44]. Developing interventions that mitigate oxidative damage and preserve vascular function may improve health outcomes, enhance quality of life, and reduce the societal burden of age-related vascular diseases [45,46].

## 2. Pathophysiology of Oxidative Stress in Vascular Aging

ROS and reactive nitrogen species (RNS) are critical mediators in cellular signaling, yet their overproduction or dysregulation has been linked to aging and vascular dysfunction [47]. These reactive molecules are primarily byproducts of normal cellular metabolism, generated in organelles such as mitochondria, peroxisomes, and the endoplasmic reticulum [48]. While low concentrations of ROS and RNS are essential for normal physiological processes, such as cell signaling, immune defense, and vascular homeostasis, their excess leads to oxidative stress, a major contributor to aging and dysfunction [49].

Aging is associated with a progressive decline in the efficiency of antioxidant defenses, leading to an imbalance between ROS/RNS production and their scavenging [50]. This imbalance results in oxidative damage to macromolecules, including lipids, proteins, and DNA [51]. Such damage accumulates over time, driving cellular senescence and the functional decline of tissues, particularly in the vascular system [52]. Mitochondrial dysfunction, a hallmark of aging, exacerbates ROS production, creating a vicious cycle of oxidative stress and energy depletion [53].

While ROS are traditionally linked to oxidative stress and cellular injury, they also serve a pivotal function in the innate immune system by contributing to the rapid and efficient elimination of invading pathogens [54]. During phagocytosis, activated neutrophils and macrophages employ a specialized enzyme complex known as NADPH oxidase to initiate an “oxidative burst”, characterized by the swift production of superoxide anions (O_2_^−^) [55]. These highly reactive anions are subsequently transformed into hydrogen peroxide (H_2_O_2_) by superoxide dismutase (SOD) [56]. In neutrophils, myeloperoxidase further converts H_2_O_2_ into hypochlorous acid (HOCl), which is considered particularly potent among these molecules due to its strong antimicrobial properties [57].

ROS also exert direct antimicrobial activity by damaging essential microbial components, such as proteins, lipids, and nucleic acids, ultimately aiding in the clearance of pathogens from the host [58]. ROS generated during the oxidative burst also function as critical signaling entities, orchestrating the recruitment and activation of additional immune cells and amplifying the local inflammatory response [54]. This underscores the delicacy of redox homeostasis, where a finely tuned equilibrium of ROS levels is required to balance protective functions against deleterious outcomes [59].

The dual role of ROS in both microbial eradication and tissue injury highlights the importance of tightly regulated oxidative mechanisms in immune defenses. Understanding the complexity of ROS involvement in innate immunity offers a broader perspective on their contribution to physiological and pathological processes, including oxidative stress and aging. This understanding emphasizes the nuanced interplay between protective and detrimental effects across various biological systems [60].

In the vascular system, ROS and RNS play dual roles [61,62]. Physiological levels of these species regulate vascular tone by modulating endothelial nitric oxide (NO) bioavailability, a key determinant of vasodilation [63]. However, in pathological conditions such as hypertension, diabetes, and atherosclerosis, excessive ROS and RNS levels reduce NO bioavailability through oxidative inactivation, leading to endothelial dysfunction [64,65]. Endothelial cells become pro-inflammatory and pro-thrombotic, contributing to vascular stiffness and impaired blood flow [66].

ROS-induced oxidative stress activates redox-sensitive transcription factors, such as NF-κB and AP-1, which upregulate the expression of pro-inflammatory cytokines and adhesion molecules [67]. This fosters leukocyte adhesion, smooth muscle cell proliferation, and vascular remodeling, which are processes implicated in age-related vascular diseases [68]. Similarly, RNS, particularly peroxynitrite (ONOO^−^), generated from the reaction of NO with superoxide, causes nitration of tyrosine residues in proteins, altering their structure and function [69]. This has been linked to impaired mitochondrial respiration and endothelial dysfunction [70].

Another significant contributor to ROS and RNS generation in aging is the NADPH oxidase (NOX) enzyme family [71]. NOX enzymes are major sources of superoxide and hydrogen peroxide in the vasculature [72]. Key enzymes such as superoxide dismutase (SOD), catalase, and glutathione peroxidase, which act as antioxidants, show reduced activity with age. This decline in enzymatic defense exacerbates oxidative stress by allowing the accumulation of ROS and RNS [73,74]. Furthermore, the uncoupling of endothelial nitric oxide synthase (eNOS) leads to the production of superoxide instead of nitric oxide, compounding endothelial dysfunction and vascular aging [75]. Unlike mitochondrial ROS production, which is a byproduct of ATP synthesis, NOX enzymes are dedicated to generating superoxide as their primary function. The age-related upregulation of NOX activity amplifies oxidative stress and promotes vascular inflammation [76,77,78]. Conversely, the inhibition of NOX has been shown to restore vascular function in experimental models of aging, highlighting its potential as a therapeutic target [79]. The NOX family comprises several isoforms with distinct tissue distributions and regulatory mechanisms [80]. During aging, the expression and activity of specific NOX isoforms, such as NOX2 and NOX4, are upregulated in endothelial and vascular smooth muscle cells, leading to chronic oxidative stress. This heightened NOX activity is implicated in the pathogenesis of hypertension, atherosclerosis, and other vascular diseases [81].

Mitochondrial dysfunction is another pivotal source of ROS in aging. The electron transport chain (ETC) within mitochondria becomes increasingly inefficient with age, resulting in electron leakage and the production of superoxide anions [82].

Mitochondrial ROS damages cellular components and disrupts mitochondrial DNA (mtDNA), further impairing the organelle’s function. The accumulation of mtDNA mutations during aging perpetuates a cycle of mitochondrial dysfunction and oxidative stress, contributing to the decline in vascular health [83].

One of the underappreciated roles of NOX enzymes is their interaction with mitochondrial ROS production. Studies suggest a crosstalk mechanism in which NOX-derived ROS can damage mitochondria, leading to the production of further ROS. This feed-forward loop intensifies oxidative stress, creating a scenario often called “ROS-induced ROS release”. Understanding this interplay could open avenues for dual-target therapies aimed at both NOX enzymes and mitochondrial dysfunction [84].

Mitochondria-targeted antioxidants have gained attention for their role in mitigating oxidative stress in aging and vascular dysfunction [85]. Compounds such as MitoQ and SkQ1 specifically accumulate within mitochondria and neutralize ROS at their source [86]. Experimental data demonstrate that these antioxidants improve endothelial function, reduce vascular stiffness, and delay the progression of age-associated vascular diseases [87]. However, their clinical efficacy and long-term safety remain areas of ongoing investigation [88].

Metabolic factors, including hyperglycemia and dyslipidemia, further influence the interplay among ROS, RNS, and vascular aging [89]. In diabetes, for example, elevated glucose levels drive excessive ROS production through mechanisms such as protein kinase C activation and advanced glycation end-product (AGE) formation [90,91]. These processes exacerbate oxidative stress and vascular damage [92]. Similarly, dyslipidemia promotes lipid peroxidation, generating reactive aldehydes like malondialdehyde, which further impair endothelial function [93,94].

Chronic inflammation, a hallmark of aging often termed “inflammaging”, is closely linked to ROS and RNS [95]. Persistent low-grade inflammation induces ROS production via immune cell activation and mitochondrial dysfunction [96]. This, in turn, perpetuates inflammatory signaling pathways, creating a feedback loop that accelerates vascular aging. Targeting this vicious cycle is a promising therapeutic avenue [97].

Caloric restriction and exercise are two non-pharmacological interventions that counteract ROS- and RNS-mediated vascular dysfunction [98]. Caloric restriction reduces mitochondrial ROS production by enhancing efficiency and biogenesis [99], while exercise upregulates antioxidant defenses and stimulates NO production, thereby improving endothelial function [100,101,102]. These lifestyle modifications not only delay aging but also mitigate vascular complications in aging populations [103].

Emerging evidence suggests that dietary polyphenols, such as resveratrol and quercetin, exhibit potent antioxidant properties that neutralize ROS and RNS [104,105]. These compounds activate endogenous antioxidant pathways, such as the nuclear factor erythroid 2-related factor 2 (Nrf2) pathway, enhancing cellular resilience to oxidative stress [106].

The gut microbiota has also emerged as a critical regulator of ROS and RNS balance in aging and vascular health [107,108]. Dysbiosis, or microbial imbalance, increases intestinal permeability and systemic inflammation, promoting oxidative stress [109]. In preclinical models, probiotic and prebiotic interventions aimed at restoring gut microbial homeostasis have shown promise in reducing vascular oxidative stress [110].

In aging-associated vascular calcification, ROS and RNS contribute to the osteogenic transformation of vascular smooth muscle cells. Oxidative stress activates signaling pathways, such as the Wnt/β-catenin and RANKL pathways, which promote calcification [111].

Cellular senescence, a state of irreversible growth arrest, is both a cause and a consequence of ROS/RNS dysregulation in aging. Senescent endothelial and vascular smooth muscle cells exhibit a senescence-associated secretory phenotype (SASP), characterized by the secretion of pro-inflammatory and pro-oxidant factors. These factors exacerbate local and systemic oxidative stress, driving vascular dysfunction [112,113,114].

## 3. Cellular Senescence in Vascular Smooth Muscle Cells and Its Role in Vascular Stiffness

Vascular smooth muscle cells (VSMCs) play a pivotal role in maintaining vascular homeostasis, mediating structural integrity, and regulating vascular tone [115]. However, with aging and various pathological insults, these cells undergo phenotypic alterations, including cellular senescence [116]. Cellular senescence is characterized by a permanent state of cell cycle arrest, resistance to apoptosis, and the acquisition of a pro-inflammatory secretory phenotype, termed the senescence-associated secretory phenotype (SASP) [117]. In the context of vascular smooth muscle cells, senescence contributes to vascular stiffness by altering extracellular matrix (ECM) dynamics, promoting inflammation, and impairing contractile function [118].

One key mechanism driving senescence in VSMCs is the accumulation of oxidative stress [119]. ROS generated by mitochondrial dysfunction, NADPH oxidases, or reduced antioxidant defenses can induce DNA damage and activate signaling pathways, such as p53/p21 and p16INK4a, leading to cell cycle arrest [120]. Senescent VSMCs secrete matrix metalloproteinases (MMPs) and pro-inflammatory cytokines, which degrade the ECM and recruit immune cells to the vascular wall. This remodeling results in a loss of vascular elasticity and an increase in stiffness [121].

Telomere shortening is another significant factor contributing to VSMC senescence [122]. With repeated cell division, telomeres—the protective caps at the ends of chromosomes—shorten, eventually triggering a DNA damage response and cellular senescence. Shortened telomeres have been implicated in aging-related vascular diseases, including atherosclerosis and hypertension. In VSMCs, telomere shortening is exacerbated by oxidative stress, further compounding the senescent phenotype [123].

Epigenetic changes, such as DNA methylation and histone modifications, also play a role in VSMC senescence. These changes alter gene expression patterns, promoting pro-inflammatory and ECM-degrading profiles. The accumulation of senescent VSMCs within the vascular wall disrupts its contractile phenotype, reducing its ability to respond to vasoactive stimuli and further contributing to vascular stiffness [124,125,126].

Increased vascular stiffness is a hallmark of aging and is associated with a host of cardiovascular diseases, including hypertension, atherosclerosis, and heart failure. The loss of vascular compliance elevates systolic blood pressure and pulse pressure, placing additional strain on the heart and contributing to end-organ damage. Targeting VSMC senescence through pharmacological or genetic interventions may offer therapeutic potential to mitigate vascular stiffness and improve cardiovascular outcomes [127,128,129].

Endothelial cells (ECs) line the inner surface of blood vessels and are critical for maintaining vascular health by regulating vasodilation, inflammation, and thrombosis [130,131]. A key function of endothelial cells is the production of NO, a potent vasodilator synthesized by eNOS. However, oxidative stress—an imbalance between the production of ROS and antioxidant defenses—can disrupt NO signaling, leading to endothelial dysfunction and vascular pathology [132].

Under physiological conditions, NO is synthesized by eNOS in response to shear stress or agonists such as acetylcholine [133]. NO diffuses to adjacent vascular smooth muscle cells, where it activates guanylyl cyclase, leading to cyclic GMP production and vasodilation [134]. Oxidative stress impairs this process through several mechanisms. Firstly, ROS directly scavenge NO to form peroxynitrite, a highly reactive and damaging molecule that reduces NO bioavailability [135]. Secondly, oxidative stress uncouples eNOS, causing it to produce superoxide instead of NO, which further exacerbates endothelial dysfunction [136].

Mitochondrial dysfunction is a significant source of ROS in endothelial cells. With aging or pathological conditions, such as diabetes and hypertension, mitochondria exhibit impaired electron transport chain function, which leads to increased ROS production. Other sources of ROS include NADPH oxidases, xanthine oxidase, and inflammatory cells that infiltrate the vascular wall [137].

The disruption of NO signaling has profound consequences for vascular health. Reduced NO bioavailability diminishes endothelial-dependent vasodilation, leading to increased vascular tone and blood pressure. Moreover, peroxynitrite and other ROS-mediated byproducts cause oxidative damage to lipids, proteins, and DNA, triggering inflammation and endothelial cell apoptosis. These processes contribute to vascular remodeling, atherosclerosis, and thrombosis [138].

Strategies to restore NO signaling and combat oxidative stress include lifestyle interventions, such as exercise and dietary antioxidants, as well as pharmacological approaches. Angiotensin-converting enzyme (ACE) inhibitors, angiotensin receptor blockers (ARBs), and statins have been shown to enhance endothelial function by reducing oxidative stress and improving NO bioavailability. Novel therapies targeting eNOS uncoupling or enhancing mitochondrial function hold promise for mitigating endothelial dysfunction and its associated vascular complications [139].

## 4. Gasotransmitters Biology and Mechanisms

Nitric oxide (NO), hydrogen sulfide (H_2_S), and carbon monoxide (CO) are commonly referred to as gasotransmitters—small, gaseous signaling molecules that play critical roles in various physiological and pathophysiological processes [140,141]. Unlike traditional neurotransmitters or hormones, these gases are membrane-permeable and can diffuse freely through cells and tissues without the need for specific transporters [142]. Their discovery revolutionized our understanding of cell signaling, as it became evident that these gases could modulate a wide array of biological activities, including vasodilation, neurotransmission, and immune responses [143].

Historically, NO was the first gas recognized for its biological functions, notably as an endothelium-derived relaxing factor that regulates vascular tone [144]. Following the footsteps of NO, both H_2_S and CO were also found to have similarly impactful roles [145]. H_2_S, once thought of solely as a toxic gas, is now known to affect processes ranging from inflammation to neuroprotection. CO, most commonly associated with poisoning from exhaust fumes, also participates in essential cellular signaling pathways, influencing mitochondrial function, anti-inflammatory responses, and vascular homeostasis [146].

A pivotal characteristic of gasotransmitters is their dual nature; they are crucial at low, regulated concentrations but can become toxic when levels rise beyond physiological thresholds [147]. Their small size, high diffusibility, and often transient half-lives allow them to reach targets rapidly and exert localized effects [40]. Moreover, crosstalk among the NO, H_2_S, and CO pathways has been increasingly documented, revealing layers of complexity in gas-mediated signaling networks. As research unfolds, these molecules continue to gain prominence not only for their physiological roles but also as potential therapeutic agents, targeting the underlying mechanisms of diseases ranging from hypertension and heart failure to neurodegenerative disorders and inflammatory conditions [148].

Gasotransmitters are synthesized through highly regulated enzymatic pathways, producing them in precise amounts at the right times [149]. NO is primarily generated by NOS, which has three isoforms: endothelial (eNOS), neuronal (nNOS), and inducible (iNOS). The endothelial and neuronal isoforms typically produce NO in low, controlled amounts for signaling, whereas the inducible isoform can produce large quantities in response to inflammatory stimuli [150]. In turn, H_2_S is synthesized mainly by cystathionine β-synthase (CBS) and cystathionine γ-lyase (CSE), enzymes that use sulfur-containing amino acids such as cysteine and homocysteine as substrates [151]. CO is formed principally via the heme oxygenase (HO) pathway, in which HO-1 (inducible) and HO-2 (constitutive) break down heme into biliverdin, free iron, and CO [152].

The regulation of these enzymes is complex and can be influenced by factors such as oxidative stress, inflammatory mediators, and signaling molecules from overlapping pathways. For instance, NO synthesis can be upregulated during inflammation, while H_2_S production may increase or decrease depending on cellular stress and disease states. CO production is often elevated in conditions of oxidative stress or tissue damage, as HO-1 is strongly induced by these insults. These interplay mechanisms help maintain cellular and physiological homeostasis [153].

Once generated, NO, H_2_S, and CO can modulate a variety of target molecules. NO commonly signals through soluble guanylyl cyclase (sGC), elevating cyclic guanosine monophosphate (cGMP) levels and influencing smooth muscle relaxation, platelet aggregation, and synaptic transmission. H_2_S targets include ion channels (particularly ATP-sensitive potassium channels) and can post-translationally modify proteins via persulfidation, affecting their structure and function. CO can also activate sGC, although its effects often center on anti-inflammatory pathways and the regulation of mitochondrial respiration. Collectively, these gasotransmitters exhibit intricate regulation and multifaceted signaling mechanisms, highlighting their importance in both health and disease [154].

Gasotransmitters are renowned for their capacity to act as potent regulators of redox balance and antioxidant defense [155]. All three molecules, NO, H_2_S, and CO, can scavenge ROS directly or enhance the activity of intrinsic antioxidant systems, such as superoxide dismutase and glutathione peroxidase [156]. For instance, NO can react with superoxide to form peroxynitrite, although under controlled conditions, this reaction can help neutralize excess free radicals [157]. Meanwhile, H_2_S has been shown to upregulate glutathione levels and suppress NADPH oxidase activity, thereby mitigating oxidative stress in various tissues [158]. CO, often through heme oxygenase-1 (HO-1) induction, promotes the generation of bilirubin, another potent antioxidant. By fine-tuning these pathways, the gasotransmitters collectively help maintain cellular homeostasis and prevent damage arising from oxidative stress, inflammation, and other deleterious processes [159].

Beyond their antioxidant properties, NO, H_2_S, and CO play an integral role in smooth muscle relaxation, endothelial repair, and anti-inflammatory signaling [160]. These actions, together with their capacity to limit inflammatory responses—through the inhibition of pro-inflammatory cytokines and suppression of leukocyte adhesion—help preserve endothelial function and facilitate tissue repair. During vascular injury, the balanced production of these gasotransmitters supports the regeneration and remodeling of damaged vessels. Consequently, their combined effects on redox balance, vascular relaxation, and inflammation underscore the immense therapeutic potential of modulating NO, H_2_S, and CO for treating cardiovascular, neurodegenerative, and inflammatory disorders [161].

The interplay among NO, H_2_S, and CO in the vasculature reveals a complex, finely tuned network that significantly impacts vascular function and overall cardiovascular health [162]. Although each gasotransmitter has distinct modes of synthesis and primary targets, mounting evidence suggests that their signaling pathways converge and influence one another, synergizing processes such as vasodilation, endothelial protection, and anti-inflammatory responses. For example, NO can upregulate the expression of cystathionine γ-lyase (CSE), an enzyme responsible for generating H_2_S, thereby enhancing the latter’s cytoprotective and vasorelaxant actions. Similarly, H_2_S has been shown to stabilize NO by reducing its degradation, thus prolonging NO-mediated signaling. CO also interacts with both NO and H_2_S pathways; while its affinity for heme groups is well-documented, CO can modulate eNOS activity and affect H_2_S production through shifts in redox balance [147,149].

## 5. Therapeutic Potential of Gasotransmitters in Vascular Aging

H_2_S, once primarily regarded as a toxic gas, is now recognized as a vital signaling molecule with significant roles in regulating oxidative stress and inflammation. One of the central mechanisms by which H_2_S exerts its protective effects is through its ability to directly and indirectly modulate ROS. For instance, H_2_S can chemically interact with oxidative species, helping to neutralize free radicals. Additionally, it enhances endogenous antioxidant capacity by upregulating enzymes such as superoxide dismutase (SOD) and glutathione peroxidase, thereby improving the cell’s overall defense against oxidative damage. This antioxidant capability is closely tied to H_2_S’s anti-inflammatory potential. In many inflammatory conditions, elevated levels of pro-inflammatory cytokines and leukocyte infiltration lead to tissue damage. H_2_S has been shown to reduce the expression of these cytokines and limit leukocyte adhesion, dampening the inflammatory cascade. By attenuating both oxidative stress and inflammation, H_2_S safeguards cellular components, particularly in vulnerable tissues like the cardiovascular and nervous systems [163,164].

A key aspect of H_2_S-mediated cytoprotection involves its influence on mitochondrial function. Mitochondria are pivotal in energy metabolism and ROS production, and H_2_S helps maintain their integrity under stress. By modulating the electron transport chain, H_2_S can optimize ATP production while reducing excessive ROS generation. It also impacts mitochondrial biogenesis and promotes the activity of antioxidant enzymes located in or near the mitochondria, ensuring that these organelles remain efficient and resilient. Another notable feature of H_2_S signaling is its involvement in persulfidation, a post-translational modification of cysteine residues in target proteins. This modification can alter protein conformation and function, often leading to enhanced enzymatic activity or improved stress resistance. Collectively, these mechanisms underline H_2_S’s multifaceted role in preserving cellular health; it neutralizes harmful radicals, bolsters antioxidant defenses, reduces the burden of inflammation, and ensures that mitochondria—central hubs of metabolic regulation—continue to function optimally [165].

The bioavailability of NO critically depends on healthy endothelial function. However, aging, oxidative stress, and pro-inflammatory states can diminish eNOS activity and increase NO degradation, thereby contributing to endothelial dysfunction. In age-related hypertension, reduced NO signaling is a prominent factor, as lower NO availability leads to increased arterial stiffness and higher systemic vascular resistance. Consequently, systolic blood pressure tends to rise in older adults [166]. Therapies aimed at restoring or enhancing NO levels—such as phosphodiesterase type 5 (PDE5) inhibitors, angiotensin-converting enzyme (ACE) inhibitors, and nitrate supplementation—help offset these effects by improving NO release and/or reducing its breakdown [167]. Lifestyle modifications, including regular exercise, dietary interventions rich in nitrate-containing vegetables (e.g., beetroot), and antioxidant supplementation, can further bolster NO’s protective role by attenuating oxidative stress [168].

In atherosclerosis, endothelial damage and plaque formation are exacerbated by insufficient NO. Low NO levels facilitate inflammatory cell infiltration and abnormal smooth muscle proliferation within the arterial wall, accelerating plaque development. By maintaining or restoring endothelial NO production, it is possible to counteract these processes and slow atherosclerotic progression. Pharmacological and non-pharmacological strategies designed to enhance NO bioactivity have emerged as promising approaches for managing age-related hypertension and mitigating atherosclerosis [169].

Long regarded as a dangerous pollutant, CO has emerged as a critically important gasotransmitter with antioxidant and anti-inflammatory properties when produced at low, physiologically relevant levels. Endogenously, CO is generated mainly through the activity of heme oxygenase (HO) enzymes, especially HO-1, which breaks down heme into biliverdin, iron, and CO [170].

Once synthesized, CO modulates various signaling pathways, conferring cytoprotective benefits by reducing oxidative damage and dampening inflammatory responses. For instance, CO can suppress the expression of pro-inflammatory cytokines, such as tumor necrosis factor-alpha (TNF-α) and interleukins, partly by inhibiting transcription factors like NF-κB. This suppression curtails the infiltration and activation of immune cells at sites of injury, thereby mitigating tissue damage and promoting a more controlled healing environment. Additionally, through its interaction with heme-containing proteins, CO can influence cellular redox states and indirectly bolster antioxidant defenses [171,172].

In vascular health, CO exerts protective effects against injury and fibrosis, processes often linked to excessive inflammation and endothelial dysfunction. By modulating the balance between vasoconstriction and vasodilation, CO helps maintain blood vessel homeostasis, similar to the role of nitric oxide (NO). Specifically, CO can activate sGC, although it is less potent than NO, to induce smooth muscle relaxation and improve blood flow [173]. This vasodilatory action, coupled with CO’s ability to reduce platelet aggregation and leukocyte adhesion, helps preserve endothelial integrity following vascular insults. Moreover, experimental models indicate that CO signaling can limit the proliferation and migration of smooth muscle cells, which are key contributors to vascular remodeling and pathological fibrosis. By mitigating excessive smooth muscle growth and collagen deposition, CO helps maintain vessel elasticity and structural stability. Collectively, these attributes underscore CO’s therapeutic promise for managing vascular injury and chronic fibrotic conditions, suggesting that targeted modulation of the HO-1/CO pathway could offer novel strategies for cardiovascular protection and repair [174].

## 6. Emerging Therapeutic Strategies

The therapeutic potential of gasotransmitters has spurred the development of various exogenous donor molecules designed to release them in controlled and physiologically relevant doses (Figure 2).

Improved application systems for direct vascular delivery of gasotransmitters focus on enhancing targeted and sustained release while minimizing systemic effects [177]. Nanoparticles and hydrogels enable controlled, site-specific delivery, responding to environmental stimuli such as pH or oxidative stress [178]. NO-releasing nanoparticles reduce endothelial dysfunction, while H_2_S-loaded hydrogels promote vascular repair in ischemic tissues [179]. Polymeric and lipid-based nanoparticles have been developed for NO delivery in cardiovascular diseases. These nanoparticles reduce endothelial dysfunction, promote angiogenesis, and prevent restenosis. An example is NO-releasing silica nanoparticles, which enhance vasodilation and improve vascular healing post-angioplasty [180].

Hydrogels are highly hydrophilic polymeric networks capable of absorbing large amounts of water, making them ideal for the localized and sustained release of gasotransmitters. They can be engineered to degrade under specific physiological conditions, ensuring a controlled-release profile [181]. These hydrogels have shown potential in promoting vascular regeneration in ischemic tissues by providing a continuous supply of H_2_S. Stimuli-responsive hydrogels release H_2_S in response to oxidative stress. For example, a hydrogel composed of partially oxidized alginate cross-linked with tetraaniline nanoparticles and engrafted with 2-aminopyridine 5-thiocarboxamide and adipose-derived stem cells as a source of H_2_S was injected into rats after myocardial infarction, resulting in improved left ventricle function and a decrease in the fibrotic area [182].

NO-loaded hydrogels have been used in wound healing applications to promote angiogenesis and reduce infection risks. They are particularly beneficial in chronic wound management and ischemic limb diseases [183]. Recent advances combine NO and H_2_S in a single hydrogel system, exploiting their synergistic effects on vascular protection and anti-inflammatory activity [184,185,186].

NO Prodrugs (e.g., NONOates): S-nitrosothiols and NONOates provide sustained NO release and are being investigated for hypertension and pulmonary hypertension treatment. These compounds have longer half-lives and reduced systemic side effects compared to conventional NO donors.

H_2_S Prodrugs (e.g., GYY4137): Unlike fast-releasing H_2_S donors like NaHS, prodrugs such as GYY4137 release H_2_S slowly over extended periods, which reduces cytotoxicity while maintaining therapeutic efficacy. They have shown promise in protecting against cardiac ischemia and promoting angiogenesis.

Carbon Monoxide-Releasing Molecules (CORMs): CORMs are designed to release CO in a controlled manner, minimizing toxicity while maximizing anti-inflammatory and cytoprotective effects. Photocontrolled CORMs release CO upon exposure to specific wavelengths of light, enabling precise temporal and spatial control.

New engineered donor molecules (Table 1) are shaping the next generation of gas-based therapies by providing more precise control over gas release and limiting off-target effects [187]. Prodrugs are inactive compounds that are metabolized into active gasotransmitters at the target site, providing a more predictable and controlled release compared to direct gas administration [188].

For H_2_S, sodium hydrosulfide (NaHS) and GYY4137 are prominent examples that can liberate H_2_S slowly or quickly, depending on the compound’s chemistry, thus allowing researchers to tailor the delivery rate for specific applications [189]. For NO, compounds like S-nitroso-N-acetylpenicillamine (SNAP) and diazeniumdiolates (NONOates) can donate NO in a sustained manner, improving the half-life and overcoming challenges associated with rapid NO breakdown [190]. Similarly, carbon monoxide-releasing molecules (CORMs) have been engineered to release CO under certain conditions, often triggered by pH, light, or temperature changes. These CORMs aim to circumvent the toxicity issues associated with inhaled CO by ensuring localized release at therapeutic concentrations [191].

Advances in nanotechnology further enhance the therapeutic potential of gasotransmitters through targeted and stimulus-responsive delivery systems. Nanocarriers—constructed from polymers, lipids, or inorganic materials—can encapsulate gas donor compounds, shielding them from premature degradation while traversing biological barriers. These nanovehicles can be functionalized with ligands that target specific tissues or cell types, thus improving drug accumulation at the target site [192].

Stimulus-responsive systems offer an added level of sophistication; under particular physiological conditions (e.g., low pH in tumor microenvironments or elevated oxidative stress in inflamed tissues), the nanoparticle can release its payload of gas donor molecules, thereby achieving spatiotemporally precise treatment. Such smart delivery technologies also minimize systemic exposure and reduce the risk of adverse effects. Together, the evolution of gasotransmitter-releasing compounds and the advent of advanced nanotechnologies hold considerable promise for leveraging the cytoprotective, anti-inflammatory, and pro-regenerative properties of H_2_S, CO, and NO in a wide range of clinical settings [193].

Efforts to harness the therapeutic potential of gasotransmitters are increasingly being integrated with antioxidant-based and mitochondrial-targeted treatments to optimize clinical outcomes. Because these gases exert potent antioxidant effects and help maintain cellular redox balance, combining them with established antioxidant therapies can amplify cytoprotective mechanisms. For example, the co-administration of H_2_S donors and agents like N-acetylcysteine (NAC) or vitamin E can strengthen the overall defense against ROS, reducing oxidative damage to lipids, proteins, and DNA [194].

Similarly, treatments directed at mitochondria—such as mitochondria-targeted antioxidants (e.g., MitoQ) or uncouplers that enhance mitochondrial efficiency—can work in tandem with gasotransmitter donors. H_2_S, NO, and CO each influence mitochondrial function by regulating respiration, ATP production, and ROS generation; thus, a combination strategy can help sustain mitochondrial integrity while maximizing energy utilization and minimizing oxidative stress. Furthermore, such regimens may bolster anti-inflammatory pathways, as both gasotransmitters and antioxidants can dampen excessive immune responses and protect tissues from inflammatory injury [195,196].

In parallel, synergy with lifestyle interventions—notably exercise and diet—underscores a promising avenue for amplifying the beneficial effects of gasotransmitter-based treatments. Aerobic exercise, for instance, stimulates endothelial function, raises endogenous NO levels, and promotes the enzymatic production of H_2_S [197]. Likewise, specific dietary patterns, such as those rich in nitrates (from leafy greens) and sulfur-containing amino acids (from cruciferous vegetables, garlic, and onions), can enhance the body’s capacity to synthesize these gasotransmitters [198]. These lifestyle factors can further improve vascular health, oxidative status, and tissue repair when combined with exogenous donor therapies. Additionally, exercise and diet can regulate body weight, lipid profiles, and glucose metabolism, thereby reducing comorbid risk factors [199].

Despite promising preclinical data on NO, H_2_S, and CO therapies, translating these findings into safe and effective clinical treatments presents considerable challenges. These small molecules have short half-lives, diffuse rapidly, and exert highly localized actions, making precise dosing difficult. Approaches such as gasotransmitter-releasing molecules or nanocarriers aim to deliver these gases in a controlled manner, yet striking the right balance between efficacy and safety remains an ongoing challenge [200]. Overly rapid release of a gas can trigger toxicity, while insufficient release may produce sub-therapeutic effects. Moreover, the body’s endogenous production of these gases and their interplay with other signaling pathways further complicate dosage regulation. Relying solely on plasma or tissue measurements may be insufficient, as local concentrations of gasotransmitters often do not accurately reflect systemic levels. Advanced drug delivery systems must incorporate real-time monitoring and feedback loops to adjust release rates [201].

Safety and efficacy issues become even more critical in older adults with age-related vascular diseases. Aging is frequently associated with endothelial dysfunction, oxidative stress, and inflammation—conditions in which gasotransmitter-based therapies could be beneficial [202]. However, older adults may also exhibit compromised organ function, polypharmacy risks, and greater susceptibility to adverse effects; thus, therapeutic doses of H_2_S donors might provoke hypotension or gastrointestinal upset [203], while NO donors could cause undesirable drops in blood pressure or rebound endothelial dysfunction [204].

Additionally, comorbidities—such as diabetes or kidney disease—can interfere with how these molecules are metabolized or distributed. Clinical trials must, therefore, carefully stratify patients based on comorbid status and medication regimens while rigorously evaluating both short- and long-term outcomes. By refining controlled-release technologies and tailoring treatments to individual patient profiles, gasotransmitter therapies could overcome current translational roadblocks and pave the way for safer, more effective management of age-related vascular diseases [205].

The transition of gasotransmitter-based therapies from preclinical studies to clinical application presents numerous challenges, particularly concerning safety, dosing, and individual variability. Uncontrolled NO release can lead to severe hypotension, limiting its clinical use, especially in patients with pre-existing cardiovascular conditions. Innovative delivery systems, such as NO-releasing nanoparticles and catheter-based infusions, are being developed to provide localized and sustained NO release, thereby reducing the risk of systemic hypotension. The narrow therapeutic window of CO also poses significant safety concerns, with high doses leading to toxicity, hypoxia, and neurological damage. The use of stimulus-responsive CORMs offers additional control by releasing CO only under specific physiological conditions (e.g., low pH or oxidative stress). Also, the metabolic response to H_2_S varies significantly among individuals, influenced by genetic polymorphisms in the key producing enzymes. This variability can impact both efficacy and safety.

Individual variability in response to gasotransmitter-based therapies is a critical factor affecting both efficacy and safety. Variations in the NOS genes—especially eNOS (NOS3)—are associated with altered NO bioavailability and vascular response. Patients with such polymorphisms may require higher NO doses or alternative therapeutic strategies to achieve optimal outcomes. H_2_S production is primarily regulated by the enzymes CSE and CBS. Polymorphisms in these genes, such as rs1047891 in CBS, can affect H_2_S synthesis, leading to variability in antioxidant defense and mitochondrial protection. Individuals with reduced H_2_S production may benefit from exogenous H_2_S donors or combination therapies targeting oxidative stress. Genetic differences in the HO-1 (HMOX1) gene, which is responsible for CO production, can alter the anti-inflammatory and cytoprotective effects of CO-based therapies. Shorter repeats in the HO-1 promoter are associated with higher enzyme activity, while longer repeats reduce CO production and may decrease therapeutic efficacy.

Beyond genetic variability, differences in diet, lifestyle, and comorbid conditions, such as diabetes or chronic kidney disease, can further influence individual responses to gasotransmitter therapies. Personalized treatment plans that incorporate genetic screening, metabolic profiling, and real-time monitoring of gasotransmitter levels may help optimize therapy and minimize adverse effects.

## 7. Discussion

The studies reviewed underscore the immense therapeutic potential of gasotransmitters in modulating vascular tone, reducing oxidative stress, and dampening inflammation. These findings align with earlier hypotheses suggesting that small, gaseous molecules exert potent cytoprotective effects through mechanisms that include direct free radical scavenging, enhancement of endogenous antioxidant enzymes, and fine-tuning of mitochondrial function [205]. Results across diverse experimental models point to the value of synergy among gasotransmitters, indicating that activating one pathway may reinforce the benefits of another. From a translational perspective, this supports combination therapies targeting multiple gasotransmitter axes, supplemented by antioxidant drugs and lifestyle interventions, to maximize efficacy in age-related vascular disorders [206].

Gasotransmitters present a compelling example of the Janus-faced nature of biological molecules [207], serving as both essential signaling agents and potential drivers of pathology. From an environmental perspective, these gases are integral to pollution dynamics, with sources such as vehicle emissions, industrial processes, and biomass burning contributing to their exogenous presence. Chronic exposure to these environmental pollutants is increasingly linked to oxidative stress, vascular dysfunction, and accelerated aging, particularly in the cardiovascular system [208].

Prolonged exposure to high levels of NO and CO disrupts endothelial function, impairs vasodilation, and exacerbates vascular stiffness [209], while excessive H_2_S from industrial leaks can disrupt mitochondrial respiration and increase inflammation [210].

From a cybernetic viewpoint, gasotransmitters operate as key nodes within biological signaling networks, facilitating rapid intracellular and intercellular communication. Their ability to freely diffuse across membranes without transporters allows them to act as universal messengers, linking environmental stimuli to physiological responses. NO, for example, targets sGC to induce vasodilation, while CO interacts with cytochrome c oxidase, and H_2_S modulates ATP-sensitive potassium channels (K_ATP) and influences protein persulfidation. Tissue levels of these gasotransmitters, typically in the nanomolar to micromolar range, reflect a dynamic balance governed by tightly controlled enzymatic systems, including nitric oxide synthases (NOS), heme oxygenases (HO), and cystathionine gamma-lyase (CSE). Disruptions in these systems, whether due to endogenous dysregulation or exogenous exposure, can shift the signaling dynamics from homeostasis toward pathology, amplifying oxidative and nitrosative stress and driving age-related vascular dysfunction. Advanced measurement techniques, such as electrochemical sensors, fluorescent probes, and mass spectrometry, have provided real-time insights into the spatiotemporal dynamics of gasotransmitter levels, further illustrating their cybernetic regulation within complex biological networks [211,212].

The duality of gasotransmitters emerges most strikingly when considering their role in feedback loops that bridge environmental and biological systems. Under physiological conditions, endogenous gasotransmitters act as finely tuned cybernetic regulators of vascular tone, inflammation, and metabolic function. However, environmental exposure to exogenous gasotransmitters, such as NO from vehicle exhaust or CO from industrial emissions, perturbs these feedback systems, overwhelming regulatory mechanisms and accelerating age-related damage. This Janus-faced behavior is particularly evident in vascular aging, where the dysregulated balance between protective signaling and oxidative stress leads to endothelial dysfunction, impaired vasodilation, and heightened susceptibility to cardiovascular diseases. By incorporating a cybernetic perspective, we can better understand the complex feedback loops and signaling networks that govern the dual roles of gasotransmitters, providing insights into strategies for mitigating their pathological effects while harnessing their therapeutic potential in addressing aging and pollution-related health challenges [213].

While NO, H_2_S, and CO have been extensively examined in the context of vascular health, a growing body of research suggests they also play significant roles in non-vascular aging processes. For instance, in the nervous system, NO and H_2_S are involved in synaptic plasticity and neurotransmission, influencing cognitive function and potentially modulating age-related neurodegenerative conditions such as Alzheimer’s and Parkinson’s diseases. H_2_S, for example, can enhance cellular defenses against oxidative stress in neurons and help maintain mitochondrial integrity—two factors closely tied to neuronal survival during aging. Meanwhile, CO has been reported to exert anti-inflammatory effects in glial cells, hinting at therapeutic potential for age-associated disorders characterized by excessive neuroinflammation [214,215].

Beyond the central nervous system, evidence points to the involvement of gasotransmitters in other tissues and organs prone to age-related decline. H_2_S has been shown to improve mitochondrial function in skeletal muscle, which may mitigate sarcopenia by preserving muscle mass and contractility [216]. Similarly, NO can influence muscle repair and regeneration by regulating satellite cells, which are critical for muscle maintenance [217]. In metabolic tissues, these gases can modulate insulin sensitivity and lipid metabolism. For instance, H_2_S and NO both regulate processes related to glucose uptake in skeletal muscle and adipose tissue, offering a potential link to age-related metabolic disorders such as type 2 diabetes [218,219].

Moreover, a crucial topic of interest is how gasotransmitters influence cellular senescence and autophagy—two cellular programs central to the aging process. Compelling findings suggest that modest increases in NO and H_2_S can foster autophagic flux, thereby preventing the buildup of damaged proteins and organelles. In contrast, dysregulated or excessive gasotransmitter production may drive oxidative stress or inflammatory signaling that accelerates tissue dysfunction. Future studies integrating advanced omics technologies, animal models of accelerated aging, and clinical research in older adults will be paramount in elucidating the precise role of gasotransmitters in non-vascular aging. Such investigations hold promise not only for deepening our scientific understanding but also for identifying novel interventions targeting the multifactorial mechanisms underlying age-related diseases [220].

Despite these advances, a broader context reveals that our current understanding of gasotransmitters is incomplete. In addition to NO, H_2_S, and CO, emerging evidence points to other candidate gasotransmitters—such as sulfur dioxide (SO_2_) and even ammonia (NH_3_)—that may have regulatory roles in vascular biology and beyond. Investigations into the enzymatic generation and signaling pathways of these lesser-studied gases are in the early stages but hold promise for uncovering novel therapeutic targets. Likewise, our knowledge of the molecular crosstalk among gasotransmitter pathways continues to expand, highlighting interactions with ion channels, transcription factors, and epigenetic modulators. This raises the exciting prospect of tailored interventions that exploit precise mechanistic links to treat conditions ranging from hypertension and atherosclerosis to neurodegeneration and cancer [221,222,223].

Future research should focus on characterizing these emerging gasotransmitters, elucidating their tissue-specific impacts, and devising targeted delivery systems to optimize therapeutic indices. Longitudinal clinical trials that incorporate well-defined patient stratification and biomarker monitoring will be essential for validating safety, efficacy, and long-term outcomes. By building on the insights gained thus far, researchers can develop novel strategies that harness the multifaceted power of gasotransmitters in preventing and managing a broad spectrum of human diseases.

As our understanding of vascular aging advances, it becomes evident that a one-size-fits-all strategy may be insufficient to address the diverse mechanisms underlying age-related endothelial dysfunction, arterial stiffness, and atherosclerosis.

Personalized medicine aims to tailor therapeutic interventions to individual patient profiles, guided by genetic, epigenetic, environmental, and lifestyle factors. This precision approach could involve genomics-based screening for polymorphisms linked to altered gasotransmitter (e.g., NO, H_2_S) production, inflammation, or oxidative stress in vascular aging. Such insights would help identify patients who are at higher risk for accelerated arterial aging and could benefit most from targeted therapies.

Beyond genetic determinants, integrating transcriptomic, proteomic, and metabolomic data can refine patient stratification by revealing molecular signatures that indicate early-stage endothelial dysfunction or heightened susceptibility to vascular injury. At the same time, advanced imaging techniques, such as pulse wave velocity (PWV) measurements and endothelial function assessments (e.g., flow-mediated dilation), can serve as non-invasive biomarkers for vascular health. These diagnostic tools allow clinicians to detect subtle age-related changes, paving the way for early interventions and more precise monitoring of therapeutic outcomes [224,225,226].

Drug regimens targeting gasotransmitters—such as NO donors, H_2_S-releasing compounds, or CO-releasing molecules—could be individually optimized based on each patient’s metabolic status and genetic predisposition. Personalized approaches might also incorporate complementary strategies like antioxidant therapy or mitochondria-targeted treatments while adjusting for comorbidities, medication interactions, and specific risk factors like hypertension, diabetes, or dyslipidemia. Moreover, lifestyle interventions—including tailored exercise programs, stress reduction techniques, and dietary modifications—are often integral to preserving vascular health and can be uniquely customized according to a patient’s baseline fitness, dietary patterns, and socio-behavioral context [227].

## 8. Conclusions

Gasotransmitters have emerged as potent modulators of vascular function, playing critical roles in mitigating oxidative stress, maintaining endothelial integrity, and slowing the progression of vascular aging. Far from simply being toxic gases, these small molecules are now recognized as essential regulators of redox homeostasis, anti-inflammatory signaling, and mitochondrial performance. Their ability to regulate vascular tone, promote endothelial repair, and reduce inflammatory damage underscores their broad influence on age-associated cardiovascular diseases such as atherosclerosis and hypertension. Experimental and clinical studies suggest that harnessing these gasotransmitters through exogenous donor compounds, genetic modulation of their biosynthetic enzymes, or lifestyle interventions that optimize their endogenous production can yield promising therapeutic outcomes. However, the therapeutic application of gasotransmitters remains in its nascent stage, with several hurdles to overcome.

Achieving precise dosing, optimizing delivery strategies, and accounting for inter-individual variability in baseline gasotransmitter levels are essential for translating research findings into clinical practice. Moreover, the synergistic or additive effects of combining NO, H_2_S, and CO in specific contexts highlight the importance of further elucidating their overlapping and distinct signaling pathways. As vascular aging does not occur in isolation but rather intersects with metabolic, neurological, and immune-related processes, future investigations should also consider the broader systemic impact of gasotransmitters.

Key areas for further research include developing more advanced delivery systems—particularly nanotechnology-based vehicles—to ensure the controlled, site-specific release of these gases, as well as integrating personalized medicine approaches that tailor therapies to individual genetic, epigenetic, and lifestyle factors. Ultimately, a deeper mechanistic understanding of how NO, H_2_S, and CO intertwine at the molecular and cellular levels will pave the way for refined interventions.

## Figures and Tables

**Figure 1 pharmaceuticals-18-00344-f001:**
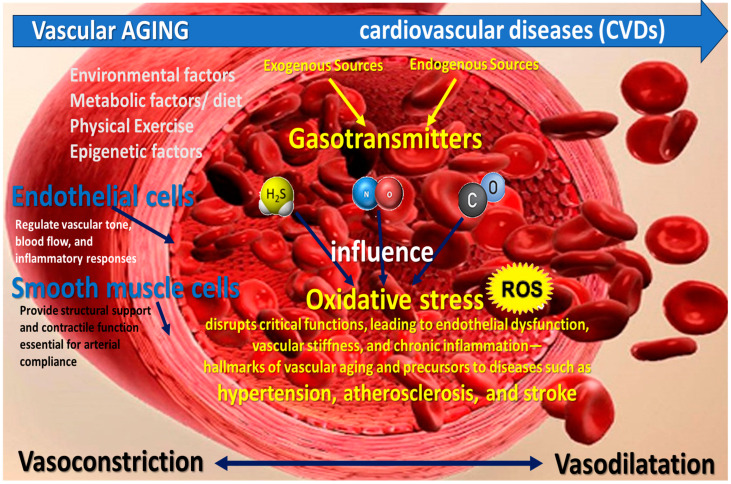
Complex interplay among vascular aging, gasotransmitters, oxidative stress, and cardiovascular diseases (CVDs). Hydrogen sulfide (H_2_S), nitric oxide (NO), and carbon monoxide (CO) influence vascular homeostasis. Aging-associated molecular changes and ROS-mediated damage accelerate vascular dysfunction and the onset of CVDs.

**Figure 2 pharmaceuticals-18-00344-f002:**
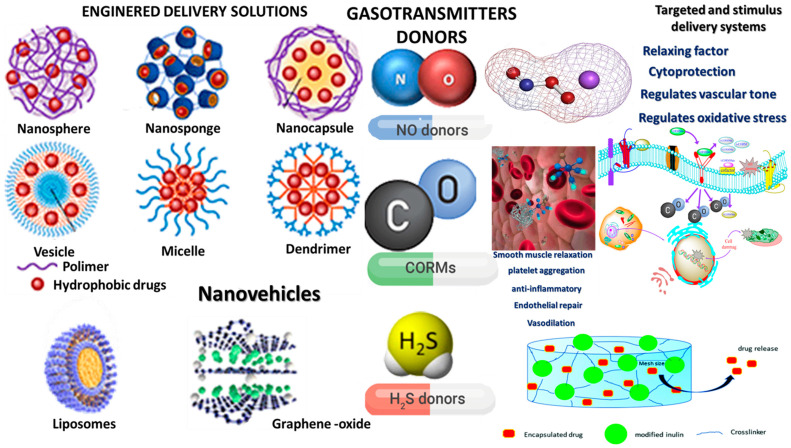
Nanovehicles designed for the targeted delivery of therapeutic agents—gas-releasing molecules, such as nitric oxide (NO), carbon monoxide (CO), and hydrogen sulfide (H_2_S) donors. These gaseous molecules act as therapeutic agents, offering diverse biological effects, including cytoprotection, regulation of vascular tone, oxidative stress mitigation, smooth muscle relaxation, platelet aggregation modulation, anti-inflammatory effects, endothelial repair, and vasodilation [175,176].

**Table 1 pharmaceuticals-18-00344-t001:** Summary of gasotransmitter therapeutic applications, mechanisms, delivery systems, clinical development status, and comparison with alternative therapies.

Gasotransmitter	Therapeutic Applications	Mechanism of Action	Delivery Systems	Clinical Development Status	Comparison with Alternative Therapies
**Nitric Oxide (NO)**	Cardiovascular diseases (hypertension, restenosis), wound healing, pulmonary hypertension	Vasodilation, inhibition of platelet aggregation, reduction of inflammation	NO-releasing nanoparticles, drug-eluting stents, hydrogels, NONOates	Phase II–III clinical trials for cardiovascular applications	Antioxidants reduce oxidative stress but do not directly improve endothelial function as NO does; NO has immediate vasodilatory effects
**Hydrogen Sulfide (H_2_S)**	Myocardial protection, ischemia-reperfusion injury, wound healing, neuroprotection	Antioxidant defense, mitochondrial protection, stimulation of angiogenesis	H_2_S-loaded nanoparticles, hydrogels, slow-releasing prodrugs (e.g., GYY4137)	Preclinical to early clinical trials	Antioxidants provide general cytoprotection, whereas H_2_S directly supports mitochondrial bioenergetics and promotes angiogenesis
**Carbon Monoxide (CO)**	Anti-inflammatory therapy, organ transplantation, ischemic stroke protection	Modulation of heme oxygenase pathway, reduction of pro-inflammatory cytokines, cytoprotection	CORMs, CO-releasing nanoparticles, photocontrolled delivery systems	Preclinical trials for inflammation and organ protection	Anti-inflammatory drugs like corticosteroids suppress inflammation but do not offer the cytoprotective effects of CO
**Combination Therapy (NO + H_2_S)**	Cardiovascular repair, chronic wound healing	Synergistic effects on vasodilation, inflammation reduction, and endothelial protection	Dual gasotransmitter hydrogels, combined nanoparticles	Preclinical studies	Combination antioxidant therapies (e.g., vitamin C and E) have limited synergy compared to the broader cellular effects of gasotransmitters
**Balneotherapy (Mineral Springs)**	Rheumatologic disorders, cardiovascular recovery, skin diseases, chronic inflammation	Natural sources of H_2_S and CO from sulfurous and carbonated waters, stimulation of antioxidant defense	Inhalation, immersion in mineral-rich baths, topical application of mineral mud	Traditional use; growing scientific evidence from clinical studies	Non-invasive and holistic, balneotherapy provides sustained exposure to low-dose gasotransmitters and minerals, complementing pharmacological approaches

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
