# Peer review of "Harnessing Gasotransmitters to Combat Age-Related Oxidative Stress in Smooth Muscle and Endothelial Cells"

_pharmaceuticals, 2025, doi:10.3390/ph18030344_

Round 1

Reviewer 1 Report

Comments and Suggestions for Authors

In their review article, Monteano and colleagues address the influence of aging on the vascular system and the potential therapeutic role of gasotransmitters.

Comments:

- Certainly not the focus of this review article. But in my opinion, at least a small section on the role of ROS in pathogen defense is missing: phagocytosis, oxidative/respiratory burst, and so on. The role of ROS in inflammation is often discussed, but this important part is missing a little...

- Fig 2.: The image is somewhat blurred and there are still remnants of larger images visible that have not been accurately cropped.

- Improved application systems for direct vascular application are being considered, what could these be?

Author Response

We want to thank the reviewer for constructive feedback, which helped us identify areas for improvement and refine our manuscript.

- Certainly not the focus of this review article. But in my opinion, at least a small section on the role of ROS in pathogen defense is missing: phagocytosis, oxidative/respiratory burst, and so on. The role of ROS in inflammation is often discussed, but this important part is missing a little...

We appreciate the reviewer’s insightful observation regarding the role of ROS in pathogen defense, particularly during phagocytosis and oxidative/respiratory bursts. The following subsection was added:

While ROS are traditionally linked to oxidative stress and cellular injury, they also serve a pivotal function in the innate immune system by contributing to the rapid and efficient elimination of invading pathogens [1]. During phagocytosis, activated neutrophils and macrophages employ a specialized enzyme complex known as NADPH oxidase to initiate an “oxidative burst,” characterized by the swift production of superoxide anions (O₂⁻) [2]. These highly reactive anions are subsequently transformed into hydrogen peroxide (H₂O₂) by superoxide dismutase (SOD) [3]. In neutrophils, myeloperoxidase further converts H₂O₂ into hypochlorous acid (HOCl), considered particularly potent among these molecules due to its strong antimicrobial properties [4].

ROS exert direct antimicrobial activity by damaging essential microbial components, such as proteins, lipids, and nucleic acids, ultimately aiding in the clearance of pathogens from the host [5]. ROS generated during the oxidative burst also function as critical signaling entities, orchestrating the recruitment and activation of additional immune cells and amplifying the local inflammatory response [1]. This underscores the delicacy of redox homeostasis, where a finely tuned equilibrium of ROS levels is required to balance protective functions against deleterious outcomes [6].

The dual role of ROS in both microbial eradication and tissue injury highlights the importance of tightly regulated oxidative mechanisms in immune defenses. Understanding the complexity of ROS involvement in innate immunity offers a broader perspective on their contribution to physiological and pathological processes, including oxidative stress and aging. It emphasizes the nuanced interplay between protective and detrimental effects across various biological systems [7].

- Fig 2.: The image is somewhat blurred and there are still remnants of larger images visible that have not been accurately cropped.

Thank you for pointing out the issue with Figure 2. We will revise the figure to enhance clarity and remove any extraneous elements to ensure accurate visual representation.

- Improved application systems for direct vascular application are being considered, what could these be?

Improved application systems for direct vascular delivery of gasotransmitters like nitric oxide (NO), hydrogen sulfide (H₂S), and carbon monoxide (CO) focus on enhancing targeted and sustained release while minimizing systemic effects. Nanoparticles and hydrogels enable controlled, site-specific delivery, responding to environmental stimuli such as pH or oxidative stress. NO-releasing nanoparticles reduce endothelial dysfunction, while H₂S-loaded hydrogels promote vascular repair in ischemic tissues. Prodrugs like NONOates and CORMs offer localized gasotransmitter release upon activation, providing therapeutic benefits without widespread exposure.

Reviewer 2 Report

Comments and Suggestions for Authors

Highlighted limitations 1. restricted focus on clinical trials 2. repetitive content 3. insufficient discussion of the risks of gasotransmitters 4. a lack of emphasis on how individual differences may affect therapeutic response. It is recommended to expand the section on clinical translation by incorporating human studies and discussing challenges in applying these therapies to provide a more detailed assessment of safety issues such as NO-induced hypotension, CO toxicity, and the metabolic effects of H₂S, to reduce redundancies for improved readability, to analyze the impact of genetic and metabolic variability on gasotransmitter response, to enhance the discussion on drug delivery strategies, and to offer a clearer comparison with alternative therapeutic options. Further refinements could include 1. making figure captions more detailed, 2. ensuring uniformity in terminology, broadening the list of abbreviations, 3. improving sentence clarity for better readability, 4. adding a comparative table summarizing the therapeutic applications of gasotransmitters, their mechanisms, and their clinical development status.

Author Response

We want to thank the reviewer for the comprehensive feedback, which has greatly facilitated our efforts to strengthen the manuscript. Below, we detail our responses to each comment:

  1. We have broadened the scope of our discussion to include additional human studies.

  2. We have closely reviewed the text for redundancies and eliminated repetitive statements to improve readability. 

  3. Recognizing the importance of safety considerations, we have included a more in-depth exploration of risks, such as NO-induced hypotension, CO toxicity, and the metabolic implications of H₂S.

  4. We agree that patient-specific factors are central to therapeutic outcomes. Consequently, our revisions incorporate a dedicated discussion on genetic and metabolic variability and how these differences might modulate the efficacy and safety of gasotransmitter therapies.

  5. We have expanded our examination of drug delivery strategies, highlighting recent innovations. 

  6. Further Refinements

    • Figure Captions: Captions have been revised to provide more comprehensive descriptions.
    • We have standardized terms throughout and expanded the abbreviation list for clarity.
    • We have improved sentence clarity for better readability
    • We have introduced a new table summarizing the therapeutic applications, modes of action, and clinical development stages of gasotransmitters, thereby facilitating direct comparison across different approaches.

Reviewer 3 Report

Comments and Suggestions for Authors

The review by Munteaunu et al. is focused on gasotransmitters, i.e. NO, H2S and CO as molecules involved in biological mechanisms, with specific attention to their role in hampering the effects of oxidative stress in smooth muscle and endothelial cells. Indees, oxidative stress, particularly caused by ageing, may impair the function of these kinds of cells, with potential serious consequences for patients' health. The Authors introduce the topic of oxidative stress, then report about the biology and mechanisms of action of the aforementioned gases and outline the biological pathways in which they are involved, pathways which might be innovative targets to prevent oxidative stress; finally, a few lines hint at the most recent therapeutic strategies related to this topic.

The review is very well written, well organized and the cited references are very numerous and relevant to the topic. I am not quite sure the topic itself is coherent with Pharmaceuticals aim and scope, because this review is focused on gasotransmitters discussed mainly in physiological terms rather than from the pharmaceutical point of view. Therefore, I would suggest the Authors, if possible, to describe in detail the few mentioned pharmaceutical strategies: molecular formulas, mechanisms of action, results, etc. Under this perspective, Fig. 2 is very generic, and should be replaced with an image illustrating any nanovehicle described in the literature employed for delivery of NO, H2S or Co compounds.

Also, several concepts, for example the role of these molecules in cytoprotection and oxidative stress mitigation, are repeated a few times: I would suggest to limit repetitions. 

What, in my opinion, is the main fault of this review is the number of self-citations: 16 self-citations (of which 11 reviews), out of 216 total citations, is a bit too much, and I rely on the opinion of the Editor with this regard.

Author Response

We sincerely thank the reviewer for the constructive feedback. We are pleased that the review is regarded as well-written, well-organized, and relevant to the topic. We acknowledge the comments regarding the coherence of the manuscript with the journal's pharmaceutical scope, and we agree that certain aspects can be adjusted to emphasize pharmaceutical strategies better.

"The topic itself is coherent with Pharmaceuticals aim and scope... However, the review is focused mainly on physiological terms rather than pharmaceutical strategies. The Authors should describe in detail the few mentioned pharmaceutical strategies: molecular formulas, mechanisms of action, results, etc."
Thank you for pointing this out. We fully agree that a stronger emphasis on pharmaceutical applications will enhance the manuscript’s alignment with the journal’s scope. We will expand the section on therapeutic strategies, providing more detailed descriptions of molecular formulas, mechanisms of action, and preclinical or clinical results for gasotransmitter-based treatments.

"Fig. 2 is very generic and should be replaced with an image illustrating any nanovehicle described in the literature employed for delivery of NO, H₂S, or CO compounds."
We appreciate this suggestion and agree that a more specific and detailed figure will improve the manuscript. 

"Several concepts, for example the role of these molecules in cytoprotection and oxidative stress mitigation, are repeated a few times. I would suggest limiting repetitions."
Response:
Thank you for this observation. We will carefully review the manuscript to reduce redundancies regarding the mechanisms of action of gasotransmitters and their role in mitigating oxidative stress.

"The main fault of this review is the number of self-citations"
We acknowledge the reviewer’s concern regarding the number of self-citations. While the cited works were included due to their relevance and contribution to the field, we reviewed and replaced some of these references with alternative studies.

Round 2

Reviewer 3 Report

Comments and Suggestions for Authors

In the original version of the paper I had highlighted the 16 self-citations of the first author, 11 of which were review articles. In their reply, the Authors state that "we acknowledge the reviewer's concern regarding the number of self-citations. While the cited works were included due to their relevance and contribution to the field, we reviewed and replaced some of these references with alternative studies".

The number of self-citations in the revised version of the manuscript is still 16, and the only relevant modification is that one of the self-references, namely former ref. 33, has been moved forward in the manuscript and is now ref. 152. I hope the Authors forgot to replace some of the references, as they stated to have done in their reply. I am not going to revise the rest of the manuscript as long as this aspect is solved.

Author Response

Dear reviewer,

Thank you for your continued diligence in assessing our manuscript. We acknowledge your concern regarding self-citations and sincerely apologize for the confusion caused by inadvertently uploading a mistaken version. After reviewing the relevance of each citation, we removed or replaced multiple references initially connected with the same statement. We have retained only those self-references essential for our manuscript. Consequently, the total number of self-citations has been reduced to seven out of 16. Thank you very much!

Round 3

Reviewer 3 Report

Comments and Suggestions for Authors

The revised version of the manuscript, now featuring an acceptable (according to my opinion) number of self-citations, has solved the highlighted critical points, and is now suitable for publication.